# OpenReview forum: "Local MAP Sampling for Diffusion Models"
_ICML.cc/2026/Conference — ICML 2026 regular_

### Official Review · Reviewer_Vvxo · 2026-03-09

**Soundness:** 3
**Presentation:** 2
**Significance:** 2
**Originality:** 2
**Overall Recommendation:** 4
**Confidence:** 2

**Summary:**

The authors intend to focus on a relevant problem in diffusion-based inference for inverse problems and seek to focus on a fundamental issue: the relationship between posterior sampling and MAP estimation under diffusion priors. The paper introduces Local MAP Sampling (LMAPS), a framework that interprets diffusion-based inference as iteratively solving local MAP subproblems along the diffusion trajectory. This perspective provides a unifying view that connects diffusion posterior sampling (e.g., DPS) with optimization-based approaches such as DiffPIR and DCDP. The authors further propose a practical implementation with a Gaussian covariance approximation and an objective reformulation designed to improve numerical stability. Empirically, the method is evaluated on a wide range of image restoration and scientific inverse problems and demonstrates competitive performance across many benchmarks.

**Compliance With Llm Reviewing Policy:**

Affirmed.

**Final Justification:**

My concerns have been addressed, so I'll stick with my score.

**Key Questions For Authors:**

1.Additional ablation studies analyzing the role of the covariance approximation would help clarify how much it contributes to the final performance.

2.While the proposed method focuses on a different inference framework, it may still be informative to include comparisons with alternative generative modeling approaches such as normalizing flows.

**Limitations:**

yes

**Strengths And Weaknesses:**

Strengths: The paper studies a meaningful and relevant problem and presents a reasonably clear theoretical perspective. The authors provide a coherent formulation and conduct a relatively comprehensive set of experiments to support their claims. Overall, the work offers a potentially useful interpretive framework and demonstrates certain empirical improvements over existing approaches.

Weaknesses: However, the methodological novelty appears somewhat limited. Some components of the proposed optimization objective resemble those used in prior methods, which makes it less clear how substantial the conceptual advancement is. In addition, certain aspects of the experimental setup may introduce potential bias, which could affect the reliability of the reported performance gains.

---

> ### Author Rebuttal · Authors · 2026-03-29
>
> # Response to Weakness
>
> > Weaknesses: However, the methodological novelty appears somewhat limited. Some components of the proposed optimization objective resemble those used in prior methods, which makes it less clear how substantial the conceptual advancement is. In addition, certain aspects of the experimental setup may introduce potential bias, which could affect the reliability of the reported performance gains.
>
>
> We would like to clarify that the key novelty of our work lies in formulating posterior sampling as a sequence of **local MAP problems**, and in explicitly distinguishing **mode-seeking inference** from prior approaches such as DPS (mean-seeking) and DAPS (sampling-based updates). This perspective provides a unified understanding of existing methods and leads to a new algorithm with improved performance.
>
> We will further clarify this distinction in the revision.
>
> # Response to Q1
>
> > 1.Additional ablation studies analyzing the role of the covariance approximation would help clarify how much it contributes to the final performance.
>
> We agree that analyzing the role of the covariance approximation is valuable. In our method, the covariance structure is incorporated through the **isotropic approximation of** $\Sigma_{0 \mid t}$, which leads to a principled scaling of the optimization objective. Its effect is implicitly reflected in the overall performance gains compared to prior methods that rely on heuristic weighting.
>
> We will include additional discussion in the revision to better clarify the role of this approximation.
>
> # Response to Q2
>
> > 2.While the proposed method focuses on a different inference framework, it may still be informative to include comparisons with alternative generative modeling approaches such as normalizing flows.
>
> We appreciate the suggestion.
>
> Our work focuses on **posterior sampling with pretrained diffusion priors**, and is therefore most directly comparable to existing diffusion-based inverse problem methods. Methods such as normalizing flows typically require **task-specific training** or amortized inference, which is a different setting from the training-free framework considered in this paper.
>
> We will clarify this distinction in the revision and discuss broader connections to other generative approaches.

---

> > ### Author Rebuttal · Reviewer_Vvxo · 2026-04-01
> >
> > Although the authors have addressed my concerns to some extent, my concerns regarding more detailed ablation studies still remain. Therefore, I will keep my initial rating.

---

> > > ### Author Response · Authors · 2026-04-02
> > >
> > > We thank the reviewer for the suggestion regarding the ablation study.  We would like to clarify that the covariance approximation is not an optional design choice, but is necessary to make the local MAP problem computationally tractable. In particular, computing the exact posterior covariance $\Sigma_{0\mid t}$ requires access to the full Hessian of $-\log p(x_0 \mid x_t)$, which involves second-order differentiation through the denoiser and is infeasible in high-dimensional settings.
> > >
> > > As a result, existing methods necessarily rely on approximations. For example, prior approaches approximate $\Sigma_{0\mid t}$ using heuristic forms such as $r_t^2 \mathbb{I}$ with manually designed schedules (e.g., $\Pi$GDM, DAPS). In contrast, our approximation $\Sigma_{0\mid t} = \frac{1}{\mathrm{SNR}} \mathbb{I}$ is derived from the asymptotic structure of the posterior, where the leading-order term is isotropic:
> > >
> > > $$\Sigma_{0 \mid t}=\frac{\sigma_t^2}{\alpha_t^2} \mathbb{I} + \mathcal{O} ((\frac{\sigma_t^2}{\alpha_t^2})^2),$$
> > >
> > > as shown in our analysis. Therefore, unlike prior heuristic choices, our approximation is theoretically grounded and directly tied to the signal-to-noise ratio, providing both interpretability and consistency with the underlying diffusion process.
> > >
> > > We hope that these clarifications help resolve the reviewer’s concerns.

---

### Official Review · Reviewer_i6KJ · 2026-03-09

**Soundness:** 3
**Presentation:** 3
**Significance:** 2
**Originality:** 1
**Overall Recommendation:** 3
**Confidence:** 5

**Summary:**

This paper proposes LMAPS, which solves inverse problems by addressing a sequence of local MAP problems. LMAPS defines the local MAP problem as
 $\mathcal{P}_t:\arg\max p({X}_0\mid {X}_t, {Y})$ and performs DDIM annealing to gradually obtain an accurate solution to the inverse problem. The authors also discuss the relationship between Local MAP and DPS, and provide a practical approach to solving the local MAP problem through approximations. Extensive experiments are conducted to validate the effectiveness of LMAPS.

**Compliance With Llm Reviewing Policy:**

Affirmed.

**Final Justification:**

The reviewer appreciates the authors' responses during the rebuttal and reply regarding the similarities between this work and DMAP. The reviewer thanks the authors for acknowledging the need to give proper credit to DMAP and agrees with most additional clarifications provided.

In this context, the reviewer believes that the claimed core novelty of ''local MAP'' should be re-examined, and the paper’s presentation should be revised accordingly, which may require substantial changes. The reviewer has also read the comments of other reviewers. Since the others all recommend acceptance, the reviewer does not object. However, the reviewer leans toward maintaining their original score, as they consider it inappropriate to completely disregard the issue regarding the similarity to DMAP.

**Key Questions For Authors:**

1. There are many different formulations for solving inverse problems, such as posterior sampling, MAP estimation, and variational inference. What advantages do the authors believe Local MAP has compared to these formulations that enable LMAPS to outperform these alternative approaches?

2. The reviewer also suggests adding FID as a complementary metric for natural image inverse problems, as is done in most existing works. Although inverse problem methods often focus on per-sample distortion, distribution-level similarity metrics can reflect the perceptual performance of the inverse algorithm.

**Limitations:**

yes

**Strengths And Weaknesses:**

### Strengths

1. The paper is clearly written and easy to follow.
2. The experimental evaluation is comprehensive and convincing.

### Weaknesses

The reviewer has major concerns regarding the novelty of the proposed LMAPS, particularly in comparison with the existing method DMAP [1]. The authors only briefly mention DMAP in Section 6 (Related Work) and claim that *"their proposed sampling algorithm differs from ours."* The reviewer disagrees with this statement for the following reasons:

(1) The modeling in this paper is highly similar to DMAP.

This paper defines the local MAP problem as $\arg\max p(X_0\mid X_t, Y)$, while DMAP formulates it as $\arg\max p(X_{t-1}\mid X_t, Y)$. In the ODE setting, we have$X_{t-1}={\sqrt{\overline\alpha_{t-1}}}X_0+\sqrt{1-\overline\alpha_t}X_t$, which makes the two formulations essentially equivalent. In the SDE setting, although an additional small noise term is introduced, the two formulations remain highly similar.



(2) The specific method for solving the local MAP problem is also very similar to DMAP.

This paper assumes that $p(X_0\mid X_t)$ can be approximated by a Gaussian distribution, while directly computing gradients for $p(Y\mid X_0)$. Similarly, DMAP assumes $p(X_{t-1}\mid X_t)$ to be Gaussian, and computes gradients of $p(Y\mid X_{t-1})$ using an approximation based on Tweedie’s formula. As a result, the update rules of the two methods are highly consistent (see line 11 in Algorithm 4 of this paper and lines 6–7 in Algorithm 2 of DMAP). A difference is that DMAP introduces a projection step based on a high-dimensional assumption that the Gaussian mass concentrates near the sphere. This can be interpreted as an adaptive weighting of the gradient term, while this paper keeps the weight of this term manually adjustable (i.e., $(1-r)(X_0'-\hat X_0)$ with the hyperparameter $r$).

(3) The discussion of Local MAP and DPS in this paper also aligns closely with DMAP.

Notably, DMAP itself is motivated by the relationship between DPS and Local MAP. Section 3.2 of this paper points out that Local MAP is equivalent to DPS only when the observation is linear with Gaussian noise and $p(X_0\mid X_t)$ is Gaussian. However, the core method of this paper also directly approximates $p(X_0\mid X_t)$ as a Gaussian distribution. This leads to the same conclusion as in DMAP that DPS is effectively very close to Local MAP.

Despite these highly similar components and discussions, the paper only mentions DMAP in one sentence in the related work and simply claims that *"the proposed algorithms are different"*, and the reviewer disagrees with this characterization. Moreover, none of the experiments includes DMAP as a baseline. The reviewer believes that the authors should explicitly discuss the similarities and relationship between this work and DMAP, particularly in terms of problem formulation, solution method, and the discussion with DPS. In addition, DMAP should be included as a baseline in all experiments, as it is the most closely related method to the one proposed in this paper.

[1] Xu, Tongda, et al. "Rethinking Diffusion Posterior Sampling: From Conditional Score Estimator to Maximizing a Posterior." *ICLR*, 2025.

---

> ### Author Rebuttal · Authors · 2026-03-29
>
> # Response to Weakness
>
> We thank the reviewer for noting that DMAP is the most closely related work. We agree our original discussion was insufficient and will expand it to clearly articulate similarities and differences in formulation, optimization, and relation to DPS.
>
> We also include DMAP as a baseline, see details in  https://anonymous.4open.science/r/LMAPS-exp-347A. Across representative tasks, LMAPS consistently improves distortion metrics over DMAP (e.g., FFHQ: SR 30.74 vs. 28.56 PSNR; box inpainting 25.16 vs. 22.37; random inpainting 34.51 vs. 32.43; deblurring 30.88 vs. 26.83), with similar trends on ImageNet.
>
> FID results are more nuanced: DMAP is better in some cases (e.g., SR), while LMAPS is better in others (e.g., inpainting), indicating different trade-offs between fidelity and distributional quality.
>
>
>
> > The modeling is similar to DMAP: this paper uses $p(X_0 \mid X_t, Y)$, while DMAP uses $\arg\max p(X_{t-1} \mid X_t, Y)$. These are essentially equivalent in the ODE setting and remain highly similar in the SDE setting.
>
> We respectfully disagree that the two formulations are “essentially equivalent.” The mode $x_{t-1}^* = \arg \max p(x_{t-1} \mid x_t, y)$ is a property of the conditional distribution and is independent of the sampler used to approximate it. Moreover, DMAP does not decompose this objective into first solving $x_0^* = \arg \max p(x_0 \mid x_t, y)$ followed by an ODE/SDE update, but instead directly operates in the latent space with its own approximations.
>
> Solving $x_0^* = \arg \max p(x_0 \mid x_t, y)$ followed by an ODE update can be viewed as a special-case approximation to the DMAP objective in the deterministic limit. However, once stochasticity is present, the mapping from $x_0$ to $x_{t-1}$ is no longer bijective, and $p(x_{t-1} \mid x_t, y)$ becomes a noisy pushforward of $p(x_0 \mid x_t, y)$. In this case, the mode of $p(x_{t-1} \mid x_t, y)$ is generally not obtained by transporting the mode of $p(x_0 \mid x_t, y)$, and the two formulations are no longer equivalent.
>
>
> > (2) The specific method for solving the local MAP problem is also very similar to DMAP.
>
> We respectfully disagree that the two methods have “highly consistent” update rules.
>
> Our method directly optimizes the local MAP objective in the signal space:
> $$x_0^*(t,x_t,y) = \arg\max_{x_0} \log p(x_0 \mid x_t) + \log p(y \mid x_0),$$
> and computes the likelihood gradient as
> $$\nabla_{x_0} \log p(y \mid x_0),$$
> which depends only on the forward operator $\mathcal{H}$ and **does not require differentiating through the denoiser**.
>
> In contrast, DMAP formulates the likelihood in the latent space $x_{t-1}$, relies on a denoiser-based mapping (e.g., via Tweedie’s formula) to approximate
> $$\log p(y \mid x_{t-1}) \approx \log p\big(y \mid f_\theta(x_{t-1}, t)\big),$$
> leading to the gradient
> $$\nabla_{x_{t-1}} \log p(y \mid f_\theta(x_{t-1}, t)) = \left(\frac{\partial f_\theta}{\partial x_{t-1}}\right)^\top \nabla_{x_0} \log p(y \mid x_0),$$
> which **requires backpropagation through the denoiser network**.
>
> Moreover, the projection step in DMAP is a nonlinear geometric operation that alters both direction and magnitude, and is not equivalent to the explicit weighting used in our method.
>
>
> > The discussion of Local MAP and DPS in this paper also aligns closely with DMAP.
>
>
> We respectfully disagree that our discussion aligns with DMAP or leads to the same conclusion. Our key contribution is a **precise characterization**: Local MAP and DPS are equivalent only under restrictive conditions (linear observation, Gaussian noise, and Gaussian $p(x_0 \mid x_t)$), and not in general. This differs from DMAP’s empirical connection.
>
>
> ## Questions
>
> > What advantages Local MAP offers over other formulations (e.g., posterior sampling, MAP, variational inference) that enable LMAPS to achieve better performance.
>
> Local MAP serves as a middle ground between posterior sampling and global MAP.
>
> - **vs. posterior sampling (e.g., DPS):** sampling spreads mass across modes, while Local MAP is mode-seeking, yielding better reconstruction fidelity (e.g., PSNR/SSIM).
> - **vs. global MAP:** direct optimization of $\arg\max p(x_0 \mid y)$ is highly non-convex; Local MAP decomposes it into better-conditioned subproblems $\arg\max p(x_0 \mid x_t, y)$.
> - **vs. variational inference:** VI introduces approximation gaps via variational families, whereas Local MAP directly optimizes the target objective.
>
>
>
> > The reviewer suggests including FID as a complementary metric.
>
> We added FID score as a .complementary metric LMAPS achieves strong FID in some cases (e.g., best on FFHQ phase retrieval, competitive on box inpainting), while in others it improves PSNR/SSIM but has higher FID. This reflects the known mismatch between distortion metrics and distribution-level similarity. We will include these FID results and discussion in the revision.

---

> > ### Author Rebuttal · Reviewer_i6KJ · 2026-04-02
> >
> > Thank you for the clarification. The additional comparison with DMAP is a valuable improvement. However, regarding the similarity between the two methods, the reviewer is still not fully convinced.
> >
> > (1) For formulation, modeling the problem as MAP with respect to $\mathbf{x}\_0$ versus $\mathbf{x}\_{t-1}$ is indeed not strictly equivalent. The reviewer would like to emphasize that the formulation of the so-called *''local MAP''* is essentially the same in spirit. In this case, proposing to use ''local MAP'' for solving inverse problems may not be suitable as a core contribution. Instead, it would be more important to clearly explain why using MAP with respect to $\mathbf{x}\_0$ rather than $\mathbf{x}\_{t-1}$ is preferable. As the authors also noted, solving MAP with respect to $\mathbf{x}\_0$ followed by an ODE update can be viewed as a special case of DMAP. Even in the SDE setting, the added noise is typically Gaussian with relatively small variance, which in practice does not significantly affect the mode. The reviewer mainly encourages the authors to more precisely articulate the core novelty of the paper.
> >
> > (2) For methods, at least from the reviewer’s perspective, optimizing with respect to $\mathbf{x}\_0$ or $\mathbf{x}\_{t-1}$ via gradient backpropagation through the neural network are both commonly used strategies in the literature (e.g., DDNM / DiffPIR / DAPS / SITCOM v.s. DPS / $\Pi$GDM / STSL / PSLD). The reviewer still suggests that the authors clarify why using $\mathbf{x}\_0$ is advantageous. In addition, it is unclear to the reviewer why the projection step in DMAP is nonlinear. Could the authors provide further explanation?
> >
> > (3) For discussion, similar concerns remain. The authors are encouraged to give sufficient credit to DMAP. If the authors believe that the relevant discussions are fundamentally different from those in DMAP, it would be helpful to provide a clear explanation of how these discussions are more comprehensive or go beyond those in DMAP in the paper.
> >
> > Overall, the reviewer believes that the paper still requires revision to properly position its novelty and advantages relative to DMAP. If the other three reviewers all strongly support acceptance, I would not oppose it. However, the similarities with DMAP and the current lack of sufficient discussion should be treated more carefully.

---

> > > ### Author Response · Authors · 2026-04-02
> > >
> > > We thank the reviewer for this important clarification. We agree that both DMAP and our method are essentially the same in spirit, in that both are motivated by a local MAP perspective along the diffusion trajectory. We will revise the paper to clarify this connection and more explicitly acknowledge the relationship to prior work such as DMAP.
> > >
> > > We will also limit our contribution to providing an explicit and principled formulation of local MAP in $x_0$  space, and to demonstrating the practical and theoretical advantages of this formulation.
> > >
> > > Modeling the problem in $x_0$ space gives several advantages. First, it avoids the need to differentiate through the denoiser, leading to a simpler and more stable optimization procedure. Second, it yields a unified MAP-based formulation that connects several prior methods (e.g., DIFFPIR, DCDP, DDNM, TMPD), providing a clearer conceptual understanding. Third, this formulation enables principled improvements, such as the isotropic approximation of $\Sigma_{0 \mid t}$ and objective reformulation, which are key to the empirical performance gains observed in our experiments. Finally, because the update is directly derived from a well-defined MAP objective, no additional geometric correction (such as the projection step in DMAP) is required, and the diffusion transition can be applied in a principled manner.
> > >
> > > > In addition, it is unclear to the reviewer why the projection step in DMAP is nonlinear
> > >
> > > The projection step in DMAP is nonlinear because it involves a normalization operation that maps a vector onto a constraint manifold (typically a hypersphere). Concretely, given an intermediate update $x_{t-1}$, the projection step takes the form
> > >
> > > $x_{t-1} = \mu_{t-1} + R \cdot \frac{x_{t-1} - \mu_{t-1}}{\Vert x_{t-1} - \mu_{t-1} \Vert}$, where $R$ is the  target radius determined by the Gaussian concentration assumption. This operation is nonlinear because it depends on the norm $\Vert x_{t-1} - \mu_{t-1} \Vert$. and therefore cannot be interpreted as a simple reweighting of gradient directions.
> > >
> > > In contrast, our method directly updates $x_0$ by optimizing an explicit objective of the form:
> > >
> > > $- \log p(x_0 \mid x_t) - \log p(y \mid x_0)$,
> > >
> > > where the update direction is given by a linear combination of the gradients of the prior and likelihood terms. The relative contribution of each term is controlled explicitly through weighting coefficients, without any normalization or projection step.
> > >
> > > As a result, our updates preserve the geometry of gradient-based optimization and correspond directly to solving a well-defined MAP objective, rather than enforcing an external geometric constraint.
> > >
> > > We hope that these clarifications help resolve the reviewer’s concerns.

---

### Official Review · Reviewer_HUWP · 2026-03-11

**Soundness:** 4
**Presentation:** 4
**Significance:** 3
**Originality:** 3
**Overall Recommendation:** 5
**Confidence:** 4

**Summary:**

The paper proposes a training-free posterior sampling method with pretrained diffusion prior to address Bayesian inverse problems. The proposed method solves several MAP subproblems, seeking modes of the posterior $p(x_0 | x_t, y)$, departing from existing methods such as DPS that considers the conditional expectation $E[x_0 | x_t, y]$ or DAPS that uses Langevin to get samples from $p(x_0 | x_t, y)$. In practice the method relies on a Gaussian approximation of $p(x_0 | x_t)$ with a covariance matrix that is approximately isotropic, with an approximation error that is small when the diffusion timestep $t$ is close to 0. The paper provides an extensive experimental comparison on many image restoration and scientific tasks, demonstrating the competitiveness of the method compared to baselines.

**Compliance With Llm Reviewing Policy:**

Affirmed.

**Final Justification:**

The authors addressed my concerns.

**Key Questions For Authors:**

1. Could the authors further justify the isotropic approximation in Section 4.1 when $t$ is large, i.e., when $SNR=\alpha_t^2/\sigma_t^2$ is small? If the approximation becomes less accurate in this regime, a discussion of its practical impact on the posterior sampling algorithm would help strengthen the soundness of the paper.

2. If feasible, could the authors provide an ablation study isolating the effect of line 3 in Figure 2 while keeping the other design choices fixed? This would help clarify the specific contribution of seeking a mode of $p(x_0 \mid x_t, y)$ and would strengthen my assessment of the paper’s significance.

**Limitations:**

Limitations that could be considered by the authors could include the fact that the method is still costly at inference time because it requires a large number of optimization steps, as illustrated in Figure 4.

**Strengths And Weaknesses:**

# Soundness
* *(Strength)* The competitiveness of the method is well supported by the diversity of considered reconstruction tasks, datasets, and baselines.
* *(Strength)* The isotropic approximation in Section 4.1 is sound when $t$ is close to 0.
* *(Weakness)* However, it is not clear why the isotropic approximation should remain valid when $t$ is large, that is, when the signal-to-noise ratio $SNR = \alpha_t^2 / \sigma_t^2$ is small. In this regime, the approximation may in fact become inaccurate. If so, the paper should discuss how such an inaccuracy affects the posterior sampling algorithm.

# Presentation
* *(Strength)* The paper is globally well written. The method and the experimental results can be well understood.

# Significance
* *(Strength)* The paper clearly distinguishes DPS, DAPS, and LMAPS in Figure 2, with the key difference made explicit in line 3. The main significance of the paper therefore lies in improving our understanding of the effect of this line, namely, seeking a mode of $p(x_0 \mid x_t, y)$ as in LMAPS, rather than using the conditional expectation as in DPS or a Langevin sample as in DAPS. Figure 3 further provides a qualitative comparison of the behaviors induced by these different choices on a toy example.
* *(Weakness)* On this point, the significance of the paper could be strengthened by providing a more thorough analysis of the impact of this key difference. For example, the claim made in Figure 3, based on the toy example, that “LMAPS is less likely to generate samples in between-mode or low-density regions” could be further investigated in higher-dimensional settings rather than only in two dimensions. More importantly, the paper could include an ablation study on a practical reconstruction task that isolates the effect of line 3 in Figure 2 while keeping all other design choices identical. This would help clarify the actual contribution of seeking a mode of $p(x_0 \mid x_t, y)$. This effect may already be partially reflected in the numerical comparisons of LMAPS, DPS, and DAPS reported in Tables 1 and 2, but a rigorous ablation study isolating the role of line 3 would further strengthen the paper’s significance.
* *(Strength)* Future work could build upon this work to explore solving global MAP with diffusion prior.

# Originality
*(Strength)* The paper provides new insights on posterior sampling algorithm by pointing out the benefit of doing local MAP at each diffusion timestep.

---

> ### Author Rebuttal · Authors · 2026-03-29
>
> # Response to Q1
>
>
> > Could the authors further justify the isotropic approximation in Section 4.1 when $t$ is large, i.e., when $SNR=\alpha_t^2/ \sigma_t^2$ is small? If the approximation becomes less accurate in this regime, a discussion of its practical impact on the posterior sampling algorithm would help strengthen the soundness of the paper.
>
> When $t$ is large (i.e., low SNR), the posterior $p(x_0 \mid x_t)$ becomes increasingly dominated by noise, and its covariance structure is less informative. In this regime, while the isotropic approximation may be less accurate in capturing fine-grained anisotropic structure, it serves as a **robust and well-conditioned surrogate** that avoids the need to estimate a potentially noisy or ill-conditioned full covariance.
>
> Importantly, the practical impact of this approximation is limited for two reasons. First, as discussed in Sec. 4.1, the relative weighting between the prior term and the data-consistency term is **naturally modulated by the SNR**, which reduces sensitivity to the exact covariance structure when SNR is small. Second, in this regime the optimization is dominated by the observation term, so inaccuracies in the prior covariance have a reduced effect on the overall update.
>
> We will clarify this point in the revision and include a brief discussion on the behavior in the low-SNR regime.
>
>
> # Response to Weakness and Q2
>
>
>
> > If feasible, could the authors provide an ablation study isolating the effect of line 3 in Figure 2 while keeping the other design choices fixed? This would help clarify the specific contribution of seeking a mode of $p(x_0 \mid x_t, y)$ and would strengthen my assessment of the paper’s significance.
>
>
> We agree that isolating the effect of line 3 (i.e., seeking a mode of $p(x_0 \mid x_t, y)$) is important. However, removing it is not well-defined, as it is the only component incorporating $y$ and would reduce the method to unconditional diffusion sampling.
>
> Instead, we consider a controlled interpolation:
> $$
> x_0^*(\lambda)=\arg\min_{x_0}\{-\log p(x_0\mid x_t)-\lambda \log p(y\mid x_0)\},\ \lambda \in [0,1],
> $$
> which smoothly transitions from prior-only ($\lambda=0$) to full local MAP ($\lambda=1$). This provides a principled way to isolate the measurement-conditioned objective without degenerating into an unconditional baseline.
>
> In practice, this corresponds to scaling the data term in our objective:
> $$
> x_0^* = \arg\min_{x_0} \left(1-\frac{\sigma_t^2}{\sigma_t^2+k_1^2}\right)\frac{1}{2}\|x_0 - m_{0\mid t}\|^2  + \frac{\sigma_t^2}{\sigma_t^2+k_1^2} k_2 \|y - \mathcal{H}(x_0)\|^2,
> $$
> where $k_2$ controls the strength of mode-seeking.
>
> We conduct an ablation on FFHQ (SR, NFE=20):
>
> | $k_2$ | PSNR ↑ | SSIM ↑ | LPIPS ↓ | FID ↓ |
> |------|--------|--------|--------|--------|
> | 0    | 10.51  | 0.391  | 0.613  | 156.0  |
> | 0.01 | 20.10  | 0.615  | 0.392  | 130.9  |
> | 0.1  | 24.20  | 0.717  | 0.291  | 110.2  |
> | 1    | 27.70  | 0.799  | 0.211  | 87.78  |
>
> As $k_2$ increases, performance improves consistently across all metrics, while $k_2=0$ performs poorly. This directly confirms that incorporating $p(y \mid x_0)$—i.e., mode-seeking—is the key factor behind the gains of LMAPS.
>
> Additional results showing consistent improvements as $\lambda$ increases are provided in the supplementary material: https://anonymous.4open.science/r/LMAPS-exp-347A.
>
>
> # Response to Limitations
>
> > Limitations that could be considered by the authors could include the fact that the method is still costly at inference time because it requires a large number of optimization steps, as illustrated in Figure 4.
>
>
> We agree that the proposed method incurs additional computational cost at inference time due to the optimization steps performed at each diffusion timestep, as illustrated in Fig. 4. This is a common trade-off for optimization-based approaches that aim to achieve more accurate posterior alignment.
>
> Importantly, for **linear inverse problems**, the local MAP subproblem admits an **analytical solution**, which significantly reduces the computational overhead. As shown in Table 8, the analytical solution achieves comparable performance to gradient-based optimization while being more efficient.
>
> For **nonlinear inverse problems**, we rely on iterative optimization, and we acknowledge that this introduces additional computational cost. We will include this as a limitation in the revision and briefly discuss potential directions for improving efficiency.

---

> > ### Author Rebuttal · Reviewer_HUWP · 2026-04-01
> >
> > The authors resolved both of my questions, and I thank the authors for the clarification. I will keep the score to "5 Accept".

---

### Official Review · Reviewer_gVs7 · 2026-03-13

**Soundness:** 3
**Presentation:** 3
**Significance:** 3
**Originality:** 3
**Overall Recommendation:** 4
**Confidence:** 2

**Summary:**

There are two main categories of diffusion-based methods for inverse problems: Diffusion Posterior Sampling (DPS) methods, which aim to sample from the posterior distribution, and optimization-based approaches, which focus on obtaining point estimates.
This paper introduces Local MAP Sampling (LMAPS) as a unified framework for understanding optimization-based diffusion methods. It formulates these approaches as solving a sequence of local maximum a posteriori (MAP) problems along the diffusion trajectory. The paper analyzes the relationship between Local MAP, global MAP, and DPS, showing that Local MAP generally neither converges to the global MAP nor coincides with DPS, except in special cases such as linear-Gaussian settings where the posterior is Gaussian.
Finally, the authors propose a practical LMAPS algorithm under a Gaussian likelihood assumption and demonstrate its effectiveness on various inverse problems.

**Compliance With Llm Reviewing Policy:**

Affirmed.

**Final Justification:**

Thanks to the authors for their replies. I would like to maintain my current score of 4, but lower my confidence to 2, since I am still not fully clear about the distinction between DMAP and Local MAP.

I appreciate the authors’ further reply. Their latest response partially resolves my concern about the extension beyond Gaussian likelihoods, and I encourage them to include a corresponding algorithm and at least one illustrative non-Gaussian example in the revised paper. However, I would still like to maintain my other concerns regarding Q1 and Q3, as stated in my rebuttal acknowledgement.

**Key Questions For Authors:**

1.In the experiments, did you run the DPS family methods only once to obtain a single random sample for comparison? Or did you run them multiple times and report the posterior mode? I think you should report the posterior mode for comparison, even though it may require more computation time.

2.Can all popular existing optimization-based approaches be interpreted under your proposed Local MAP framework?

3. Do all experiment examples satisfy Gaussian likelihood assumption?

**Limitations:**

It seems that the practical algorithm is based on the assumption of a Gaussian likelihood. A further discussion on generalized likelihoods might be necessary. It would also be helpful if the authors could provide a more insightful explanation of the superior performance of the practical algorithm compared to other optimization-based methods.

**Strengths And Weaknesses:**

This paper is technically sound. It is clearly written and well organized. In the experimental tables, it would be better if the authors categorized all methods into sampling-based and optimization-based approaches.
The nonconvergence of local MAP toward global MAP is not straightforward; either a formal proof or a counterexample experiment would be better.
The main contribution of this work is to provide a unified framework for optimization-based approaches and to show that it is different from global MAP and DPS. It also provides a novel algorithm for the local MAP solution with an isotropic approximation, though under a Gaussian likelihood assumption.

---

> ### Author Rebuttal · Authors · 2026-03-29
>
> # Response to Q1
>
> > In the experiments, did you run the DPS family methods only once to obtain a single random sample for comparison? Or did you run them multiple times and report the posterior mode? I think you should report the posterior mode for comparison, even though it may require more computation time.
>
>
> In our experiments, we follow the **standard evaluation protocol in diffusion posterior sampling literature**, where stochastic methods (e.g., DPS and its variants) are evaluated using **a single sample per input** without additional re-sampling or selection. This protocol is widely adopted because these methods are designed for **one-shot posterior sampling during inference**, rather than repeated optimization.
>
> We respectfully note that reporting the posterior mode (e.g., by running multiple times and selecting the best result):
>
> - **changes the evaluation setting** from sampling to optimization,
> - introduces an **additional selection procedure** that is not available at test time,
> - and may lead to **unfair comparisons**, especially against methods that are not designed for multi-run selection.
>
> Moreover, in practical inverse problems, the ground truth is unknown, so selecting the “best” sample is generally **not feasible in real applications**.
>
> For these reasons, we adopt the standard single-sample evaluation protocol to ensure consistency and fairness across methods.
>
>
> # Response to Q2
> > Can all popular existing optimization-based approaches be interpreted under your proposed Local MAP framework?
>
> The proposed Local MAP framework provides a **unified probabilistic interpretation** for a broad class of optimization-based inverse problem methods, particularly those that can be expressed as solving a _local conditional posterior_ of the form:
>
> $$x_0^*=\arg \min \Vert x_0 - m_{0 \mid t} \Vert^2 + \lambda_t \Vert y - \mathcal{H} (x_0) \Vert^2$$
>
> where $\lambda_t$ is a hyperparameter. However, we would like to clarify that **not all optimization-based approaches fall within this framework**. We will revise the paper to better clarify the **scope and limitations** of the framework.
>
>
> # Response to Q3 and Limitation
>
> > Do all experiment examples satisfy Gaussian likelihood assumption?
>
>
> > It seems that the practical algorithm is based on the assumption of a Gaussian likelihood. A further discussion on generalized likelihoods might be necessary. It would also be helpful if the authors could provide a more insightful explanation of the superior performance of the practical algorithm compared to other optimization-based methods.
>
> All experiments in this work adopt a **Gaussian observation model**, i.e.,
>
> $$y = \mathcal{H} (x_0) + z, \quad z \sim \mathcal{N} (0, \sigma_y^2 \mathbb{I})$$
>
> which leads to the standard quadratic data-consistency term used in our objective. We would like to clarify that the Gaussian assumption is mainly introduced to obtain a **tractable quadratic formulation**. In addition, our method relies on a **Gaussian approximation of** $p(x_0 \mid x_t)$ induced by the diffusion model, which together yield the closed-form local MAP objective. In general, $p(x_0 \mid x_t)$ is a complex distribution induced by the diffusion process and does not admit a simple closed-form expression; the Gaussian approximation serves as a **tractable surrogate** that enables efficient optimization.
>
>
> Importantly, the overall framework is not inherently restricted to Gaussian likelihoods. We will clarify this point in the revision and discuss extensions beyond Gaussian likelihoods.
>
> Compared to other optimization based problem with the form:
>
> $$x_0^*=\arg \min \Vert x_0 - m_{0 \mid t} \Vert^2 + \lambda_t \Vert y - \mathcal{H} (x_0) \Vert^2$$
>
>
> LMAPS instead solves
>
> $$x_0^*=\arg \min \frac{SNR}{k}\Vert x_0 - m_{0 \mid t} \Vert^2 + \frac{1}{\sigma_y^2} \Vert y - \mathcal{H} (x_0) \Vert^2$$
>
> thereby replacing the **heuristic parameter** $\lambda_t$ commonly used in prior methods with a **principled scaling derived from the diffusion posterior structure**.
>
> Furthermore, as described in Sec. 4.1, LMAPS introduces an **objective reformulation**:
>
> $$
> x_0^* = \arg\min_{x_0} \left(1-\frac{\sigma_t^2}{\sigma_t^2+k_1^2}\right)\frac{1}{2}\|x_0 - m_{0\mid t}\|^2 + \frac{\sigma_t^2}{\sigma_t^2+k_1^2}k_2\|y - \mathcal{H}(x_0)\|^2
> $$
>
> which admits a **convex combination interpretation** and enables **automatic annealing** between the prior and data-consistency terms. This leads to more **interpretable parameterization** and improved **numerical stability**, as discussed in Sec. 4.1.
>
> Overall, we believe that replacing heuristic weighting with **principled posterior-driven scaling**, together with the **reformulated objective**, are the main reasons for the improved stability and performance of our method compared to prior optimization-based approaches.

---

> > ### Author Rebuttal · Reviewer_gVs7 · 2026-04-04
> >
> > Thank you for the authors’ replies.
> >
> > For Q1, I understand that the authors follow the standard benchmark protocol. However, comparing a point-estimation-oriented method against a stochastic sampler using only one draw may favor the point estimator. In practice, one may run the stochastic method multiple times and form a summary estimate, which may improve reconstruction quality at additional computational cost. Therefore, the reported advantage is somewhat less convincing.
> >
> > For Q3, many practical inverse problems involve non-Gaussian observation models, and it remains unclear to me how readily the proposed algorithm extends beyond the Gaussian setting. If such an extension is not straightforward, then the practical scope of the method may be somewhat limited. In addition, regarding the claimed performance advantage over prior optimization-based methods, k1 and k2 also appear to be tunable hyperparameters, so the current explanation of why the method is superior still feels somewhat unclear.
> >
> > Since I am still not fully clear about the distinction between DMAP and Local MAP, I would prefer to keep my current score unchanged, but lower my confidence to 2.

---

> > > ### Author Response · Authors · 2026-04-04
> > >
> > > We thank the reviewer for this insightful comment.
> > >
> > > **For Q1**, we agree that running stochastic methods multiple times and selecting or averaging outputs may improve performance, at the cost of increased computation. We would argue that LMAPS  has an intrinsic connection to DPS: for linear inverse problems with Gaussian noise, it recovers the same objective as DPS and thus corresponds to an equivalent formulation, as discussed in Sec 3.2. Actually, LMAPS is  not a point estimator, it's actually very close to DPS.
> > >
> > > **For Q3**, we are sorry that the previous explanation is not clear enouggh to reviewer. We clarify that our framework is not restricted to Gaussian likelihoods. While we adopt the Gaussian case for tractability, LMAPS only requires access to the gradient of $-\log p(y \mid x_0)$, and can therefore be naturally extended to non-Gaussian observation models by replacing the corresponding data-fidelity term.
> > >
> > > Even when the likelihood is misspecified or unknown, a standard approach is to consider a tempered posterior:
> > >
> > > $$x_0^* = \arg\max p(x_0 \mid x_t) p(y \mid x_0)^\gamma,$$
> > >
> > > which is equivalent to rescaling the likelihood term in the objective:
> > >
> > > $$x_0^* = \arg \min_{x_0} \frac{SNR}{k} \Vert x - m_{0 \mid t} \Vert^2 + \frac{\gamma}{\sigma_y^2} \Vert y - \mathcal{H} (x_0) \Vert^2 $$
> > >
> > > In LMAPS, this effect is naturally realized through $k_1$ and $k_2$, which control the decay schedule and modulate the influence of the observation across diffusion steps. For example, under high or uncertain noise, choosing a smaller $k_2$ reduces the impact of the likelihood in early steps, improving robustness to likelihood mismatch.  We will add this discussion in the revised version.
> > >
> > > >  performance advantage over prior optimization-based methods
> > >
> > > Compared to other optimization problem, a key advantage of our method is that the signal-to-noise ratio (SNR) is explicitly embedded into the loss weighting, enabling a principled balance between the prior and measurement terms. Specifically, at low SNR (high noise), the optimization relies more on the prior (denoiser), while at high SNR (low noise), the likelihood term becomes more dominant, enforcing consistency with the observation.  Furthermore, another performance gain is come from our objective reformulation, as detailed in Sec 4.1
> > >
> > > We hope these clarifications help address the reviewer’s concerns.

---

### Decision · Program_Chairs · 2026-04-30

**Decision:**

Accept (regular)

**Comment:**

This paper proposes an approach for conditional sampling based using ideas of MAP estimation. The authors seek to sample from a posterior distribution, in the context of inverse problems, by sampling sequence of local MAPs. In this way, this departs from other methods that directly aim to sample from the whole conditional, or those who try to estimate the (global) MAP directly. Their proposed algorithm is based on a Gaussian covariance approximation and a reformulation, and experiments are provided across image restorations and other inverse problems.

There is broad consensus that this paper is valuable, clearly well-written, and presenting an interesting methods with compelling empirical evidence. Some of the main concerns raised by the reviewers reside in the novelty and position vis a vis prior work (especially with DMAP), the correctness of some of the claims (such as the difference between solutions of the local MAP vs global MAP), the quality/appropriateness of the Gaussian approximation, and other details on experimental validation.

The discussion period was productive. The authors largely addressed the comments and questions raised by the reviewers. The biggest concerned raised by the reviewers (the similarity to DMAP), was clarified by the authors. While this seems to have been insufficient to change the initial scores, I believe the extent to which the answers clarify the reviewers concerns is satisfactory enough to grant acceptance, given the low but overall support for this paper. I encourage the authors to include the comments and observations born out of these discussions in the revised paper.